# PROMPTAGATOR🐊: FEW-SHOT DENSE RETRIEVAL FROM 8 EXAMPLES

**Zhuyun Dai**[*][†]**, Vincent Y. Zhao**[*][†]**, Ji Ma**[*][†]**, Yi Luan**[*][†]**, Jianmo Ni, Jing Lu, Anton Bakalov, Kelvin Guu, Keith B. Hall and Ming-Wei Chang**[†]
Google Research
{zhuyundai, vzhao, maji, luanyi, mingweichang}@google.com
[*]equal contributions  [†]corresponding authors

## ABSTRACT

Much recent research on information retrieval has focused on how to transfer from one task (typically with abundant supervised data) to various other retrieval tasks where supervision is limited, with the implicit assumption that it is possible to generalize from one task to all the rest. However, this overlooks the fact that there are many diverse and unique retrieval problems, each targeting different search intents, queries, and search domains. In this paper, we suggest to work on *Few-shot Dense Retrieval*, a setting where each task comes with a short description and a few examples. To address this, we introduce Prompt-based Query Generation for Retrieval (PROMPTAGATOR 🐊): for each task, we feed the few-shot examples to a large language model (LLM) and prompt it to behave as a task-specific query generator. Using this, we can synthetically generate a large number of relevant queries for any document, yielding abundant data for training task-specific retrievers — with no reliance on traditional resources such as Natural Questions (Kwiatkowski et al., 2019) or MS MARCO (Nguyen et al., 2016). Surprisingly, PROMPTAGATOR using only 8 annotated examples enables efficient dual encoder retrievers to outperform computationally more expensive models trained on MS MARCO such as ColBERT v2 (Santhanam et al., 2022) by more than 1.2 points nDCG@10 on average on 11 retrieval sets. Further training standard-size rerankers using the *same* generated data yields another 5.0 points nDCG@10 improvement. Our studies show that synthetic query generation can be far more effective than previously observed, especially when a small amount of task-specific knowledge is given.

## 1 INTRODUCTION

Significant progress has been made on neural retrieval models such as dual encoders, which can search over a large collection of documents containing millions to billions of passages (Yih et al., 2011; Lee et al., 2019; Karpukhin et al., 2020). However, Thakur et al. (2021) recently proposed the BEIR heterogeneous retrieval benchmark, and showed that it is still difficult for neural retrievers to perform well on a wide variety of retrieval tasks that lack dedicated training data. To address this problem, many previous approaches focus on transferring knowledge from high-resource question answering (QA) datasets such as MS MARCO (Nguyen et al., 2016), and propose architectures that possess good inductive biases, such as models that allow fine-grained token-level interaction (e.g., ColBERT (Khattab & Zaharia, 2020; Santhanam et al., 2022) and SPLADE (Formal et al., 2021)) which often come with higher inference cost. Data augmentation via synthetic query generation has previously been explored (Ma et al., 2021; Shakeri et al., 2020), but these question generators are learned from high-resource QA datasets, and often cannot generalize well to new retrieval tasks.

We argue that it is hard to anticipate models based on one or two QA datasets to perform well across all retrieval tasks. First, different retrieval tasks have very different *search intents*; in other words, different definitions of "relevance". For example, consider Figure 1(a): both Dbpedia-Entity (Hasibi et al., 2017) and FEVER (Thorne et al., 2018) are tasks to retrieve documents from Wikipedia. However, Dbpedia-Entity is a task to retrieve entities that are mentioned in the query, while FEVER is a task to find evidence that either supports or refutes a given statement. Which

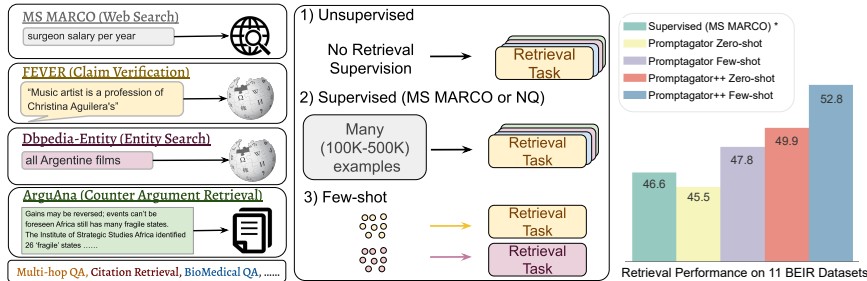

Figure 1: Few-shot retrieval with PROMPTAGATOR. **Left (a)**: Retrieval tasks from BEIR differ in query distribution, retrieval corpus, and search intents. **Middle (b)**: Most prior work uses supervised setting (2) which trains model on a large QA retrieval datasets and transfer to other retrieval tasks. **Right (c)**: Few-shot PROMPTAGATOR performance. Average nDCG@10 on 11 datasets from BEIR from our PROMPTAGATOR models and previously MS MARCO-supervised models (SPLADE v2).

document is relevant to the query can be very different from one task to another task even if they share the same domain. Moreover, different tasks have distinct distributions of queries even when their search intents are similar. For example, queries in HotpotQA (Yang et al., 2018) are long compositional questions, while queries in FiQA (Maia et al., 2018) are short financial questions.

Motivated by these observations, we advocate to work on the setting of *Few-shot Retrieval* for diverse retrieval tasks (§2), where each task comes with a short description and a few annotated examples to clearly illustrate the search intent. To address this challenge, we propose Prompt-based Query Generation for Retrieval (PROMPTAGATOR) (§3): for each new retrieval task, we feed the few-shot examples to a large language model (LLM) such as FLAN[1] (Wei et al., 2022a) and prompt it to perform doc-to-query generation. Importantly, the few-shot examples ensure that we capture the specific search intent of that task. Using this query generator, we can synthetically generate a large number of relevant queries for any document, yielding abundant data for training any retriever, including highly efficient dual encoder models.

We find that our few-shot LLM query generator can produce good queries without any fine-tuning (§3.1). In fact, as shown in Figure 1(b), our synthetically generated data is strong enough to completely forego using annotated query-document pairs from traditional high-resource datasets such as Natural Questions (Kwiatkowski et al., 2019) or MS MARCO (Nguyen et al., 2016).

While PROMPTAGATOR is not the first application of LLMs for retrieval, prior work did not explore task-specific few-shot adaptation, and often came with high inference cost. Neelakantan et al. (2022) proposes to use GPT-3 (Brown et al., 2020) in dual encoders. However, their embedding dimension is 12k, which makes the search index footprint and inference cost prohibitively high for many applications. Sachan et al. (2022) and Bonifacio et al. (2022) prompt LLMs for question generation, but did not explore the idea of using task-specific few-shot prompts for rapid task adaptation.[2] They also focus primarily on models that rerank top retrievals from an existing retriever, rather than directly adapting the underlying retriever which must efficiently search over millions or billions of documents.

To summarize, the contributions of the paper are as follows:

- We highlight previously overlooked differences across retrieval tasks (e.g., search intent and query distribution), and propose a Few-Shot Retrieval evaluation for the BEIR dataset.

- We propose PROMPTAGATOR, a simple recipe for few-shot retrieval by prompting an LLM to generate synthetic task-specific training data. For the first time, we can train fully neural retrievers and rerankers solely based on a few supervised examples.

- Our results show that, surprisingly, PROMPTAGATOR with two-to-eight examples produces significantly better retrievers than recent models trained on MS MARCO or NQ that have over 500k human annotated examples (Figure 1(c)) and utilize more expensive architectures: PROMPTAGATOR outperforms ColBERT v2 and SPLADE v2 on 11 retrieval tasks we tested, while reranking boosts results by another 5 points on standard retrieval evaluation metric.

---

[1]FLAN is a LLM that is not trained on any document retrieval or document-to-query generation tasks.

[2]InPars (Bonifacio et al., 2022) used the *same* few-shot prompt constructed from MS MARCO to generate reranker data for multiple tasks, so no task-specific prompt is used.

## 2 Few-shot Retrieval Task

In this section, we first introduce the definition of a few-shot retrieval task and discuss the differences among tasks. We then propose a new Few-Shot Retrieval setting for the BEIR benchmark.

### 2.1 Retrieval Task

Given a large corpus, a retrieval model is responsible for finding the documents that are most relevant to a provided query $q$ according to a pre-defined notion of relevance. Formally, a retrieval task is:

$$T = \{\mathcal{D}, \mathcal{Q}, \mathcal{I}\},$$

where $\mathcal{D} = \{d_1, d_2, ..., d_n\}$ is a large corpus of documents for retrieval, $\mathcal{Q}$ is a query distribution, and $\mathcal{I}$ is the underlying search intent for the task. Depending on the task, $\mathcal{D}$ can be any document collection, such as the web or Wikipedia. $\mathcal{Q}$ also varies across tasks, e.g., short keyword search queries, questions, arguments, etc. If $\mathcal{I}(q, d) = 1$, it means the search intent of $q$ has been satisfied by the document $d$. For example, in a question answering task $\mathcal{I}_{\text{QA}}(q, d) = 1$ if $d$ answers $q$. For the same $(q, d)$ pair, relevance may be either 1 or 0 depending on the search intent. For example, some argument retrieval tasks only seek to retrieve supporting arguments, while others aim to retrieve *counterarguments*.

In this work, we assume a target retrieval corpus $\mathcal{D}_\mathcal{T}$ is given, but the amount of annotated query-document pairs for the new task is limited. Most prior research efforts focused on adapting retrievers to a new corpus $\mathcal{D}_\mathcal{T}$, but didn't fully account for divergence in queries $\mathcal{Q}_\mathcal{T}$ or intents $\mathcal{I}_\mathcal{T}$. Next, we explore how a search intent can be expressed with a short description and very few examples.

### 2.2 Few-shot BEIR Setting

Intuitively, a person can understand a retrieval task by reading a short prompt and going over a few examples. In this work, we ask if *a few (8 or fewer)* examples are sufficient to learn a task-specific retriever. To facilitate our study and future research on few-shot retrieval, we define a new few-shot retrieval evaluation built upon the BEIR heterogeneous retrieval benchmark (Thakur et al., 2021).

BEIR has 18 information retrieval datasets across 9 domains, including *Bio-Medical*, *Finance*, *News*, *Twitter*, *Wikipedia*, *StackExchange*, *Quora*, *Scientific*, and *Misc*. These datasets also cover a diverse range of search intents: QA retrieval (question-to-document), duplicate question discovery (question-to-question), fact checking (claim-to-document), etc. Following Santhanam et al. (2022) and Formal et al. (2021), we narrow our focus to the publicly-available datasets in BEIR. The original BEIR evaluation used a zero-shot setup, where no queries or relevant query-document pairs from the evaluation datasets can be used for training.

We relax BEIR to the few-shot setting by *randomly* taking a few (2 to 8) in-domain relevant query-document examples as task-specific supervision — in realistic applications, this number of examples is almost always possible to obtain. The examples are sampled from the development set when it is available. For BEIR tasks which only have a test set, we use samples from the test data. To be fair when evaluating our models, we always mark these test-set examples as 'failed': we remove the documents from our retrieved results when computing metrics, even if they are correctly retrieved (the worst possible outcome). The prompts and few-shot examples will be released to the public.

## 3 Promptagator

The key idea of Promptagator is to transform a few examples into many more examples by prompting an LLM to generate more data, instead of using them to train a retriever directly.

Promptagator consists of three components: prompt-based query generation, consistency filtering, and retriever training. During prompt-based query generation, a task-specific prompt will be combined with a large language model to produce relevant queries for all documents in $\mathcal{D}_T$. Then, a filtering step cleans the generated data based on round-trip consistency. Surprisingly, we found that a retriever trained only on our synthetic data can be used to filter the synthetic data. Finally, a retriever (in this paper, dual encoders) and a cross attention reranker are trained based on the filtered data. Figure 5 in Appendix shows the overall procedure.

### 3.1 Prompt-based Query Generation

In this first step, we feed our task-specific few-shot examples into a large language model (LLM) and prompt it to perform document-to-query generation. More precisely, let $\{(q_i, d_i)\}^k$ be the $k$ few-shot examples, where each example is a query ($q_i \sim \mathcal{Q}_T$) and a document relevant to that query ($d_i \in \mathcal{D}_T$) according to the target task $T$ ($\mathcal{I}_T(q_i, d_i) = 1$).

Following FLAN (Wei et al., 2022a), we instruction-prompt the LLM with the following string prefix:

$$e_{doc}(d_i) \diamond e_{query}(q_1) \diamond \ldots \diamond e_{doc}(d_k) \diamond e_{query}(q_k) \diamond e_{doc}(d),$$

where $\diamond$ is a separator token, $e_{doc}(d)$ and $e_{query}(q)$ are task-specific document and query descriptions respectively, and $d$ is a new document presented at inference time. Using the ArguAna task as an example, we set $e_{doc}(d) =$ "`Argument:{d}`" and $e_{query} =$ "`Counter Argument:{q}`" to inform the LLM to generate counterarguments [3]. The LLM is expected to generate $e_{query}(\hat{q})$. We consider it a generation failure if the query description does not precede the actual query; otherwise, we accept $\hat{q}$ and form a synthetic relevant example $(\hat{q}, d)$.

Running the prompt on all documents from $\mathcal{D}_T$, we obtain a large set of synthetic $(\hat{q}, d)$ examples, amplifying the information from a few examples into a large synthetic dataset whose query distribution is similar to the true task distribution $\mathcal{Q}_T$ and whose query-document pairs convey the true search intent $\mathcal{I}_T$. This form of few-shot data extrapolation is similar to Lee et al. (2021).

We use FLAN (Wei et al., 2022a) as our LLM, and refer to our query generator as $p_{\text{FLAN}}(q|d)$. FLAN is trained on a collection of tasks described via instructions and was shown to have good zero/few-shot performance on unseen tasks. We use the 137B parameter checkpoint. During prompt engineering, we use at most 8 examples, and reduce the number if they exceed the input length limit of FLAN. We also manually truncate individual queries and documents in the examples if they are too long. We randomly sample up to 1 million documents from each corpus and generate 8 questions per document using sampling-based decoding with temperature 0.7.

### 3.2 Round-trip filtering generated data

We employ round-trip filtering (Alberti et al., 2019; Lewis et al., 2021) to improve the quality of our synthetic data. The main intuition is that for any synthetic query $\hat{q}$ generated from passage $d$, a good $\hat{q}$ should also retrieve its original passage $d$. In other words, the original $d$ should have high probability under some retriever, $p(d|\hat{q})$ (the reverse direction of query generation). If not, then we filter out $\hat{q}$.

Round-trip filtering has been very effective for synthetic question generation on QA tasks. However, these techniques typically rely on a question-answering model for the reverse direction filter. Since not all retrieval tasks resemble question-answering, this will not suffice in our setting.

Instead, we train an initial retriever from the *unfiltered* synthetic data, and then use it to filter the synthetic data. This works surprisingly well over the different search intents observed in BEIR. More precisely, given a synthetic query-document pair $(\hat{q}, d)$, we use the initial retriever to predict the most relevant passages for $\hat{q}$. We keep $\hat{q}$ only when $d$ occurs among the Top-$K$ passages returned by the retriever. We show this filter substantially reduces the number of synthetic queries and significantly improves retrieval performance. In Appendix F, we provide more insight into why this can work by viewing our synthetic queries as latent variables.

### 3.3 Few-shot Promptagator Retriever

Our synthetically generated data allows training task-specific neutral retrievers for tasks where in-domain fine-tuning is challenging due to data scarcity. In this work, we use a standard dual encoder retrieval architecture and we propose a simple pretrain-finetune recipe.

We pretrain our retriever on C4 with the independent cropping task from Contriever (Izacard et al., 2022a), where we treat two random crops from the same document as an artificial positive (query, document) pair and train with a cross-entropy loss over in-batch random negatives. Next, we fine-tune the dual encoder on $(\hat{q}, d)$ pairs from our prompt-based query generation, again with in-batch random negatives. After training for a set number of epochs, we apply round-trip filtering on our synthetic

---

[3]The full set of descriptions used in our prompts can be found in Table 5 in the Appendix.

| | Training Recipe | | Retrieval Architecture | | | | QGen |
|---|---|---|---|---|---|---|---|
| | Retrieval Supervision | Cross-Attn Distillation | Retriever | Token-level Retrieval | # Reranking Doc. | Serving Model Size | Model |
| Contriever | NA | | self | | | 110M | |
| GTR-XXL | MS MARCO (500K) | | self | | | 6B | |
| Splade v2 | MS MARCO (500K) | ✓ | self | ✓ | | 110M | |
| ColBERT v2 | MS MARCO (500K) | ✓ | self | ✓ | | 110M | |
| GenQ | MS MARCO (500K) | ✓ | self | | | 110M | T5 (MS MARCO) |
| GPL | MS MARCO (500K) | ✓ | self | | | 110M | T5 (MS MARCO) |
| MonoT5 | MS MARCO (500K) | | BM25 | ✓ | 1000 | 3B | |
| InPars | Few (3 from MS MARCO) | | BM25 | ✓ | 1000 | 3B | GPT-3 |
| UPR | NA | | Contriever | | 1000 | 110M+3B | T0* |
| PROMPTAGATOR | Few (0-8) | | self | | | 110M | FLAN |
| PROMPTAGATOR++ | Few (0-8) | | PROMPTAGATOR | | 200 | 110M+125M | FLAN |

Table 1: Comparison of different retrieval frameworks. Our serving models are just a 110M-size dual encoder PROMPTAGATOR and a 125M-size reranker PROMPTAGATOR++, as good quality generated data allows simple models/pipeline to achieve strong performance.[4] InPars's few-shot examples are from MS MARCO and is task-independent. See text for more details for UPR's QGen model[5].

data as described in §3.2 using this initial dual encoder, and then continue to fine-tune the dual encoder on the filtered data.

We also propose PROMPTAGATOR++, a reranker trained on the same synthetic data generated from our prompt-based QGen, which refines the retrieved candidates using a slower but more accurate cross-attention model. We train the reranker using a cross-entropy loss with 31 sampled negatives from top 200 passages retrieved by the PROMPTAGATOR retriever, which approximates the inference time distribution (reranking top 200 from the retriever).

**Zero-shot PROMPTAGATOR** Our prompt-based query generation can run in a zero-shot manner, where we universally apply the following prompt irrespective of the target task: '`{d} Read the passage and generate a query.`'. Here `{d}` denotes the document text. Training retrievers and rerankers on this data leads to zero-shot PROMPTAGATOR and zero-shot PROMPTAGATOR++. This recipe serves as a baseline to show the benefits of adapting the few-shot prompt to the target task.

## 3.4 COMPARISON WITH PRIOR METHODS

Table 1 compares the setting of PROMPTAGATOR to some recently proposed approaches. Several dimensions of our recipe are simpler: our dual encoder does not employ *hard negative mining*, *distillation from a cross-attention teacher* or *token-level retrieval*. Also, our 125M parameter reranker is smaller than most other rerankers. We aim to show that even simpler and smaller architectures can achieve excellent results if trained with synthetic data that has been few-shot adapted (§4.3). Compared to InPars (Bonifacio et al., 2022) and UPR (Sachan et al., 2022), our approach employs *task-specific* few-shot adaption, while InPars and UPR's prompts are task-independent and thus bear the same limitation of previous query generation approach (Ma et al., 2021; Wang et al., 2022). Another key difference is that prior works focused on reranking, while we enable few-shot learning for both reranking and retrieval.

## 4 EXPERIMENTS

We evaluate PROMPTAGATOR on the BEIR benchmark. We then dive deeper into the results through ablation studies and qualitative analysis.

## 4.1 IMPLEMENTATION

The original FLAN training set overlapped with 2 datasets in BEIR: NQ and Quora[6]. Most existing systems use MS MARCO for fully supervised learning and therefore do not report few or zero-

---

[5] Serving model size is shown here. FLAN ( 137B) is used for query generation, not for serving.

[5] UPR uses T0 query generation directly for reranking rather than synthetic data augmentation.

[6] Regarding NQ, FLAN is only trained on the question-to-answer task and never observes the question-passage supervision needed for retrieval training. We study the effects of NQ and Quora in FLAN in §4.3

| | arg | touché | covid | nfc | hotpot | dbp | climate | fever | scifact | scidocs | fiqa | AVG. |
|---|---|---|---|---|---|---|---|---|---|---|---|---|
| | | | | | | Retriever | | | | | | |
| *Unsupervised* | | | | | | | | | | | | |
| BM25 | 31.5 | **36.7** | 65.6 | 32.5 | 60.3 | 31.3 | 21.3 | 75.3 | 66.5 | 15.8 | 23.6 | 41.8 |
| Contriever | 37.9 | 19.3 | 27.4 | 31.7 | 48.1 | 29.2 | 15.5 | 68.2 | 64.9 | 14.9 | 24.5 | 34.7 |
| *Supervised by MS MARCO* | | | | | | | | | | | | |
| GTR-XXL | 54.0 | 25.6 | 50.1 | 34.2 | 59.9 | 40.8 | **26.7** | 74.0 | 66.2 | 16.1 | **46.7** | 44.9 |
| SPLADE v2 | 47.9 | 27.2 | 71.0 | 33.4 | **68.4** | 43.5 | 23.5 | **78.6** | 69.3 | 15.8 | 33.6 | 46.6 |
| ColBERT v2 | 46.3 | 26.3 | 73.8 | 33.8 | 66.7 | **44.6** | 17.6 | 78.5 | **69.3** | 15.4 | 35.6 | 46.2 |
| GenQ | 49.3 | 18.2 | 61.9 | 31.9 | 53.4 | 32.8 | 17.5 | 66.9 | 64.4 | 14.3 | 30.8 | 40.1 |
| GPL | 55.7 | 25.5 | 70.0 | **34.5** | 58.2 | 38.4 | 23.5 | 75.9 | 67.4 | 16.9 | 34.4 | 45.5 |
| **PROMPTAGATOR** (110M) | | | | | | | | | | | | |
| Zero-shot | 53.8 | 26.6 | 72.7 | **33.4** | 60.4 | 36.4 | 21.4 | 76.2 | 62.3 | 16.3 | 40.4 | 45.5 |
| Few-shot | **59.4** | 34.5 | **75.6** | 33.4 | 61.4 | 38.0 | 16.8 (24.0*) | 77.0 | 65.0 | **18.4** | 46.2 | **47.8** |
| | | | | | | Retriever + Reranker | | | | | | |
| *Unsupervised* | | | | | | | | | | | | |
| UPR (3B) | 50.3 | 21.3 | 60.4 | 33.3 | 72.2 | 33.8 | 9.5 | 57.3 | 69.6 | 17.3 | 45.0 | 42.7 |
| InPars (3B) | – | – | 78.4 | – | – | – | – | – | – | – | – | – |
| *Supervised by MS MARCO* | | | | | | | | | | | | |
| monoT5 (220M) | 13.2 | 27.7 | 77.8 | 35.7 | 69.5 | 41.9 | 24.5 | 80.2 | 73.6 | 16.5 | 41.4 | 45.6 |
| monoT5 (3B) | 28.8 | 20.0 | **79.5** | 38.4 | **75.9** | **47.8** | **28.0** | 85.0 | **77.7** | 19.7 | 51.4 | 51.1 |
| **PROMPTAGATOR++** (110M + 125M) | | | | | | | | | | | | |
| Zero-shot | 52.1 | 27.8 | 76.0 | 36.0 | 71.2 | 41.3 | 22.6 | 83.8 | 73.2 | 19.1 | 45.9 | 49.9 |
| Few-shot | **63.0** | **38.1** | 76.2 | 37.0 | 73.6 | 43.4 | 20.3 (24.1*) | **86.6** | 73.1 | **20.1** | 49.4 | **52.8** |

Table 2: **Main Results.** nDCG@10 on BEIR. **Retriever Comparisons (Upper Half):** Among the various kind of retrievers, both zero-shot and few-shot PROMPTAGATOR produce strong results. **Retriever+Reranker Comparisons (Lower Half):** In the scenario where speed is not a concern, reranker is often used. We train PROMPTAGATOR++ use the *same* generated data and get significant improvement. See text for more details for Climate-FEVER and Webis-Touché2020.[7]

shot results on MS MARCO. Therefore we exclude MS MARCO, NQ and Quora from our main evaluations. We report nDCG@10, the standard retrieval evaluation metric on BEIR.

For PROMPTAGATOR's query generation, we sample questions from FLAN with a temperature of 0.7. For round-trip filtering, we find that setting filtering threshold $K$ to 1 gives the best results on MS MARCO and thus use 1 for all BEIR datasets, We implement PROMPTAGATOR's dual encoders following GTR (Ni et al., 2021).To ensure efficiency, we use the T5-base encoder architecture consisting of 110M parameters. For PROMPTAGATOR++ reranking models, we initialize from a T5-base version 1.1 encoder checkpoint which has 125M parameters . More details of the reranker implementation can be found in Appendix C. At inference time, we rerank the top 200 candidates retrieved from the PROMPTAGATOR dual encoder.

We mostly follow the hyperparameters used in Ni et al. (2021). By default, we use batch size 6k; however, some of the corpora in BEIR contain only a few thousand documents, making multiple relevant documents appear in the same batch, which interacts negatively with our in-batch softmax loss. To address this issue, we split all datasets into three groups based on corpus size: small (<50k), medium (50k-500k) and large (>500k). For dual encoder training, we use 128 batch size for small datasets and 6k for others. We finetune for 5k steps for large datasets and 1k for others. For ranking models, we use batch size 64 for all datasets and finetune large datasets for 20k steps, 5k for others.

## 4.2 MAIN RESULTS

Table 2 shows the experimental results. We first notice that zero-shot PROMPTAGATOR already serves as a strong baseline, comparing favorably to other retrieval baselines trained on $\mathcal{O}(100K)$ examples from MS MARCO. Nonetheless, few-shot PROMPTAGATOR markedly improves upon zero-shot PROMPTAGATOR, increasing average nDCG@10 by over 2 points, which highlights the impact of few-shot learning. Few-shot PROMPTAGATOR, despite having a simple training procedure and model architecture, outperforms strong baselines such as GenQ (Thakur et al., 2021) and GPL (Wang et al., 2022) which also use query generation to augment training data, as well as ColBERT v2 (Santhanam

---

[7]Climate-FEVER's relevant query-document pairs in BEIR are not well-defined (§4.3), so we tried FEVER prompt on Climate-FEVER.The results are reported in (), but they are not used for computing the average. Webis-Touché2020 has an updated version; we used the original version.

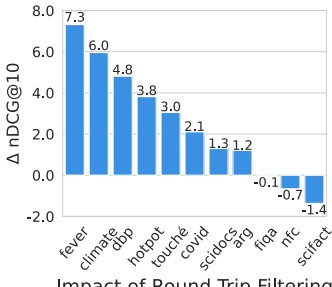 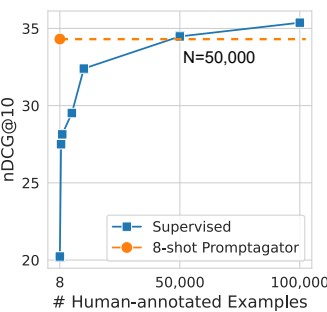 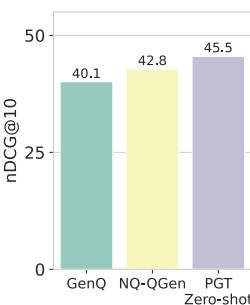

Figure 2: **Left (a)**. Delta in nDCG@10 between few-shot PROMPTAGATOR with and without filtering. **Middle (b)**: Comparing the effect of the generated data versus the number of supervised data on MS MARCO. PROMPTAGATOR with 8 examples can catch up with 50k labeled examples, when simple dual encoders are used. **Right (c)**: Ablation on query generation model. GenQ is a prior system from Thakur et al. (2021), while NQ-QGen is our in-house NQ-trained T5 query generation model. Other than the generated data, NQ-QGen and PROMPTAGATOR uses the same hyper parameters.

et al., 2022) and SPLADE v2 (Formal et al., 2021) which rely on token level interaction architectures and distillation recipes.

Our reranker PROMPTAGATOR++ boosts performance by another 5 points on nDCG@10. It significantly outperforms UPR (Sachan et al., 2022) whose reranker uses T0 (Sanh et al., 2022), an instruction tuned LLM similar to FLAN. It also outperforms monoT5-3B (Nogueira et al., 2020), which achieved previous state-of-the-art reranking performance on BEIR in a recent study (Rosa et al., 2022). Note that most of these reranker approaches use a large 3B parameter model for better generalization, while PROMPTAGATOR++ uses a standard 125M reranker.

Comparing few-shot PROMPTAGATOR to baselines, the biggest improvement is on Webis-Touché2020 (touché), followed by ArguAna (arg) . Webis-Touché2020's goal is to retrieve documents for a *controversial* topic, e.g., *"should felons who have completed their sentence be allowed to vote?"*. ArguAna's goal is to find the *counter-arguments* that oppose the input argument, and the input arguments are often several-sentence long. Both tasks are extremely different from traditional QA retrieval data that other models use, which are dominated by factoid questions. On the other hand, few-shot PROMPTAGATOR can successfully adapt to this task with a few examples.

### 4.3 ANALYSIS

**Impact of round-trip filtering.** In Figure 2(a), we show quality differences between few-shot PROMPTAGATOR with and without filtering. Filtering improves performance on 8 out of 11 datasets and leads to 2.5 points improvement on average, demonstrating the effectiveness of our filtering strategy. Nonetheless, filtering hurts model quality on NFCorpus and SciFact. These are the smallest datasets in terms of generated queries and may indicate overfitting of our retrievers.

We find that the majority of filtered examples are either queries that are too generic that match many documents, or queries that contain additional terms that are irrelevant to the document. Examples are in Fig. 6 in the Appendix. There are also cases where high quality data was incorrectly removed. We suspect that designing dynamic filtering thresholds would help, and leave it to future exploration.

**Can generated queries replace human annotated queries?** In Figure 2(b), we evaluate 8-shot PROMPTAGATOR on MS MARCO, comparing it against dual encoders trained on MS MARCO's supervised data. Note that we did not add other components, to make the comparison simple. We chose MS MARCO as there are enough labeled data for this task and neither FLAN nor our models are trained on MS MARCO examples. The results show that eight examples plus an LLM can replace a significant portion of supervised examples.

**How does PROMPTAGATOR compare to other query generation approaches?** Figure 2(c) compares zero-shot PROMPTAGATOR to two other query generation approaches: GenQ is prior system from Thakur et al. (2021) using a MS MARCO trained T5 query generation model, and NQ-QGen

|                      | arg  | touché | covid | nfc  | hotpot | dbp  | climate  | fever | scifact | scidocs | fiqa | AVG. |
|----------------------|------|--------|-------|------|--------|------|----------|-------|---------|---------|------|------|
| FLAN original        | 59.4 | 34.5   | 75.6  | 33.4 | 61.4   | 38.0 | (24.0*)  | 77.0  | 65.0    | 18.4    | 46.2 | 48.5 |
| FLAN w/o NQ and Quora| 58.8 | 33.3   | 70.2  | 33.7 | 61.7   | 34.4 | (23.5*)  | 76.2  | 63.8    | 18.3    | 43.0 | 47.0 |

Table 3: Impact of different FLAN versions. This study uses Fever prompt for Climate Fever (§4.3) .

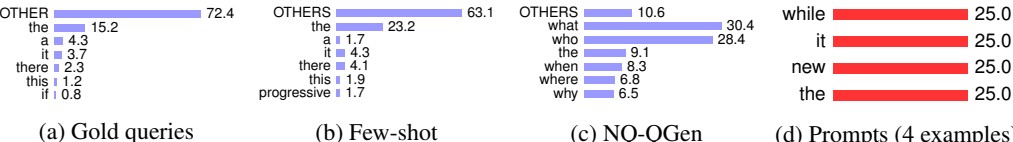

(a) Gold queries      (b) Few-shot      (c) NQ-QGen      (d) Prompts (4 examples)

Figure 3: Top first word distribution on queries generated from different models in the ArguAna dataset. **Left (a)(b)(c)**: Compare gold queries (a) and generated queries (b)(c). Queries generated by few-shot models has closer distribution to the gold queries, while the NQ-QGen queries are mostly questions. **Right (d)**: The few shot FLAN can generate diverse queries even though there are only 4 examples in the prompt. Statistics of more datasets are available in the Appendix (Figure 4).

is our in-house T5 QGen model finetuned on NQ. The figure shows the advantages of zero-shot PROMPTAGATOR, outperforming both baselines by large margins. Importantly, NQ-QGen uses the same filtering, dual-encoder training, batch sizes and training steps as PROMPTAGATOR, providing a faircomparison of query generators. This indicates that the main contributing factor to PROMPTAGATOR is better queries from prompting an LLM, not the specific training recipe or hyperparameters.

**Does few-shot always improve over zero-shot?** As shown in Table 2, few-shot PROMPTAGATOR almost always outperforms zero-shot PROMPTAGATOR except for Climate-FEVER. The original Climate-FEVER dataset uses one of three tags to annotate a query-document pair, namely "supports", "refutes", or "not enough info". However, BEIR treats all these three annotations as relevant, which is problematic. Using query-document pairs annotated "not enough info" in our prompt could be detrimental to generation quality. Therefore, we tried switching to FEVER's few-shot prompt, as the two datasets share same corpus and similar search intents. With the better annotated examples, few-shot PROMPTAGATOR indeed surpass zero-shot. This result provides some evidence that low quality few-shot examples negatively affect PROMPTAGATOR.

**Impact of FLAN Versions** FLAN was trained on a collection of datasets which have some overlap with BEIR; specifically, it includes Natural Questions (NQ) and Quora. It was not trained on query-document pairs from NQ or Quora; however, in order to determine whether the inclusion of this data biased the results on the final retrieval evaluation, we designed an additional ablation experiment. Following the original FLAN recipe (Wei et al., 2022a), we trained an additional LLM excluding both the NQ and Quora datasets. Table 4 shows the results. While the accuracy drops slightly, the overall performance still outperform prior retrievers.

**Qualitative Analysis** In order to understand the advantages of few-shot PROMPTAGATOR, we analyze the distribution of queries generated by different query generation methods for ArguAna in Figure 3. To easily see the differences, for each distribution we plot the histogram of each query's *first word*. Note that the distribution of few-shot PROMPTAGATOR (Fig. 3b) is much closer to the real distribution (Fig. 3a) while the NQ-QGen (Fig. 3c) mostly generated questions even when the queries in this task are generally arguments, not questions. More examples are showcased in Table 6 in the Appendix.

**Use few-shot examples directly** To study the effect of using the few-shot examples directly, we conduct a study of fine-tuning task-specific models using the GTR dual encoder (110M) (Ni et al., 2021) with the few-shot examples in appendix A. As expected, it does not provide a large amount of impact for GTR base, where the average nDCG decreases from 40.4 to 38.7.

## 5 RELATED WORK

**Neural Retrieval and Reranking Models** The majority of neural retrievers today employ a *dual encoder* architecture that encodes queries and documents independently into dense vectors and

retrieves documents using maximum inner product search (MIPS). Recent research has primarily focused on the following aspects: developing better pre-training tasks (Lee et al., 2019; Chang et al., 2020; Izacard et al., 2022a; Gao & Callan, 2021; Oguz et al., 2022), improving contrastive negatives (Qu et al., 2021; Xiong et al., 2021; Lu et al., 2021), and improving generalization across different domains (Thakur et al., 2021; Ren et al., 2022).

Although dual encoders enable fast retrieval, their expressivity is limited due to the fact that their score is just a dot-product between a query vector and a document vector. A common solution is to use a cross-attention model to rerank retrieved candidates (Nogueira & Cho, 2019; Nogueira et al., 2020), as cross-attention rerankers can explicitly model the interaction between query and document tokens. Distilling cross-attention models into dual encoders has been effective in closing the gap between the two (Hofstätter et al., 2020; Ren et al., 2021; Reddi et al., 2021; Zhang et al., 2022).

**Neural Retrieval with Fine-Grained Interactions** An alternative for bridging dense retrievers and cross-attention models is to allow some amount of fine-grained query-document interactions in the retriever. Humeau et al. (2020) and Luan et al. (2021) represent and retrieve queries and documents with multiple vectors instead of a single vector. ColBERT (Khattab & Zaharia, 2020), COIL (Gao et al., 2021a) and SPLADE (Formal et al., 2021) take this further by using token-level interactions between queries and documents. Because these models are not just modeling a dot product, MIPS algorithms cannot be used directly. Hence, these models usually have much higher inference/serving cost compared to dual encoders.

**Prompt-based Query Generation** The idea of using prompted LLMs for query generation has previously been proposed for improving retrieval reranking. UPR (Sachan et al., 2022) proposed to use prompted LLMs to rerank passages directly. InPars (Bonifacio et al., 2022) is probably the most closely related work to ours. They proposed to use few-shot prompting with GPT-3 to generate synthetic data for training a T5-based reranker. Though InPars was tested on multiple retrieval datasets, they used a task-independent prompt constructed from MS MARCO and did not explore task-specific few-shot learning. They also focused exclusively on reranking, whereas we also address full-scale retrieval.

**Few-shot Learning** pre-trained LLMs have significantly advanced few-shot learning, thanks to prompting strategies such as in-context learning and instruction-prompting (Brown et al., 2020; Wei et al., 2022b). Some approaches fine-tune LLMs specifically for few-shot learning (Schick & Schütze, 2021a;b;c; Gao et al., 2021b; Logan IV et al., 2022; Izacard et al., 2022b) while others do not (Brown et al., 2020; Bonifacio et al., 2022). We use LLMs as few-shot data generators — this form of data augmentation via few-shot example extrapolation is similar to Lee et al. (2021).

## 6  CONCLUSION AND DISCUSSIONS

In this paper, we present PROMPTAGATOR, a novel approach to few-shot retrieval. We showed that it is possible to create task-specific retrievers and rerankers with only a few annotated examples. The few-shot examples, amplified by prompt-based LLM query generation, simplifies the complexity of training neural retrievers for new tasks and leads to promising performance gains. It hopefully inspires future research towards generalizable retrieval systems that can seamlessly and efficiently adapt to many tasks.

While we demonstrate that LLM-based query generation can be very effective, many questions remain. One of the key issue requiring further investigation is on the generated data efficiency. We have not yet explored exactly how many query-document pairs are needed for each task, or how to use these generated examples more efficiently. Another issue is the sensitivity of the final retriever's performance with respect to the prompt. Finally, we would like to draw a connection from PROMPTAGATOR to distillation, as the final dual encoders indirectly "learn" from the LLM. Analyzing the headroom and understanding how we can better transfer knowledge from LLMs to retrievers would be a critical topic for the future.

As mentioned in §6, PROMPTAGATOR can be viewed as distilling LLM to standard-sized dual encoders via prompt-based query generation. While the distillation process is computationally expensive, it significantly reduces cost for inference.

ACKNOWLEDGEMENTS

We thank Kenton Lee, Tom Kwiatkowski, and Daniel Gillick for technical discussion and providing feedback on our manuscript. We thank Alex Salcianu for developing a bulk inference pipeline for large language models.

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

| | arg | touché | covid | nfc | hotpot | dbp | climate | fever | scifact | scidocs | fiqa | AVG. |
|---|---|---|---|---|---|---|---|---|---|---|---|---|
| GTR-base | 51.1 | 20.5 | 53.9 | 30.8 | 53.5 | 34.7 | (24.1*) | 66.0 | 60.0 | 14.9 | 34.9 | 40.4 |
| GTR-base with 8 examples | 51.9 | 24.1 | 56.2 | 30.5 | 23.4 | 35.4 | (25.4*) | 68.2 | 60.1 | 14.8 | 35.3 | 38.7 |

Table 4: Directly using eight examples might not improve the results. To showcase this, we fine-tuned a GTR-based model with the same few-shot examples used for tuning.

## COMPUTE USAGE AND ENVIRONMENTAL IMPACT

We used the 137B FLAN, which is based on LaMDA (Thoppilan et al., 2022). LaMDA was pre-trained on a large corpus consisting of 1.56T words, costing 451 MWh energy and 25.2 tCO2e carbon footprint. In PROMPTAGATOR, we generated 29.23M queries * 2 prompts = 58.46M queries, for a total of 610M words.

## A    FINE-TUNING GTR WITH FEW-SHOT EXAMPLES

We study the effect of directly adding few-shot (no more than 8 examples) on top of a 110M dual encoder, GTR-base Ni et al. (2021). To maximum the utilities of the 8 examples, we associated each positive example with 32 random negative examples from the target corpus to construct a batch, and ran 50 steps of fine-tuning for each task. As expected, the 8 examples did not provide a large amount of impact for GTR base, where the average ndcg go from 40.4 to 38.7 after fine-tuning using few-shot examples.

## B    ANALYSIS ON PROMPTS

Table 5 shows the list of prompt templates on different BEIR datasets. In order to further analysis the difference between zero-shot and few-shot prompts, we compare the few-shot and zero-shot generated queries given the same paragraph, randomly sampled from three datasets in Table 6. We observe that in general, the few-shot generated queries are closer to the original queries, while zero-shot queries are mostly questions. For example, in the ArguAna dataset, the few-shot queries are in general longer and more claim-like. In contrary, the zero-shot queries are most short question-like queries. Interestingly, for the HotpotQA dataset, even though both few-shot and zero-shot queries are generating questions-like queries, few-shot queries sometimes generate multi-hop questions, while zero-shot mostly generates single-hop questions. We further conduct first word distribution across different generation models for all datasets in Figure 4.

| Dataset | Prompt |
|---|---|
| ArguAna | `0 Argument:  passage X 1 Counter argument:  query X` |
| FiQA | `0 passage X 1 query X` |
| HotpotQA | `0 Evidence:  passage X 1 Vexed question:  query X` |
| DBPedia-Entity | `0 entity:  passage X 1 query:  query X` |
| NFCorpus | `0 Article:  passage X 1 Query:  query X` |
| Touché-2020 | `0 passage X 1 Debate:  query X` |
| TREC-Covid | `0 passage X 1 Question:  query X` |
| SciFact | `0 passage X 1 Finding:  query X` |
| SCIDOCS | `0 passage X 1 The passage is about query X` |
| FEVER | `0 passage X 1 Is it true that query X` |

Table 5:  Prompt template for each dataset.

| Paragraph | Few-shot | Zero-shot | Analysis |
|---|---|---|---|
| ArguAna: Banning skin whitening on such a basis also requires the acceptance of the racial overtones. Some form of tan is popular in many societies of people of European ethnic origins – that is not a racial matter, it is more based on economic social perceptions (that of holidays to warmer climates). Ascribing a racial element to everything to do with skin tone is at best a lazy analysis. Irrespective of issues of race and perceptions of ethnic origins, and its intersection with beauty standards, some people will be given advantages in life due to their appearance. Banning a certain form of cosmetic, even if it can have some racial and ethnic undertones, won't change that. | Query 1: Banning skin whitening on such a basis also requires the acceptance of the racial overtones. However, the racial overtones are simply a side effect of the product being used incorrectly in order to attain a lighter skin tone than they are naturally born with. The product itself is not racist, but rather the people who use it. For example, a person of Asian descent may use a skin whitening cream to attain a lighter skin tone so that they may fit into a more white-dominant society. The cream itself is not racist, but rather the fact that the person has used it to attain a lighter skin tone so that they may fit into a more white-dominant society. | Query 1: What did they try to ban? Query 2: Is skin whitening racist? Query 3: Banning skin whitening on such a basis also requires the acceptance of the racial overtones. Some form of tan is popular in many societies of people of European ethnic origins – that is not a racial matter, it is more based on economic social perceptions (that of holidays to warmer climates). Query 4: What are some of the racial overtones in skin-whitening? Query 5: Are tans a sign of affluence? | In ArguAna dataset, the few-shot examples are more **statement-like** and are **longer** than zero-shot. |
| Touché-2020: Psychologically, diet-wise and academically. Housewives can be both educated and uneducated but the fact that they spend more time with their children means that they put in a greater effort and since they are not preoccupied with their job(read: cellphone); they pay the close attention needed to keep their children out of trouble. Think of housewives as inspectors in your home... | Query 1: Housewives are better parents Query 2: Should housewives be considered better parents? | Query 1: diet-wise and academically. Query 2: People want to be relaxed all the time(At home AND at work) that is why all the research on work-life balance. Query 3: has shown that home-makers are more relaxed, less stressed, healthier, happier - the list goes on. | The few-shot generates **argument-like** queries that are more **controversial**, while zero-shot generate random statements that sometimes even have grammar problems. |
| HotpotQA: The Yung See San Fong House, or Ÿoung's Home in the Heart of the Hills, is a house in Los Gatos in the U.S. state of California. It is a combination of Asian decorative motifs and pagoda roofs together with Western massing and layout. It was completed in 1917 for Ruth Comfort Mitchell Young, a writer, and her husband, Sanborn Young, a gentleman farmer, conservationist and later California State Senator. | Query 1: Which author was the husband of Ruth Comfort Mitchell Young? Query 2: Which is known as the Yount-Lee House Query 3: Who is the architect of the Yung See San Fong House? Query 4: The Yung See San Fong House, or Ÿoung's Home in the Heart of the Hills, is a house in which U.S. state? | Query 1: Yung See San Fong House was completed in 1917 for Ruth Comfort Mitchell Young, a writer, and her husband, Sanborn Young. Query 2: How old is the Yung See San Fong House? Query 3: Yung See San Fong House was completed in 1917 for Ruth Comfort Mitchell Young, a writer, and her husband, Sanborn Young. | Few-shot examples sometimes will create multihop questions indicated in blue, which rarely happens in zero-shot examples. |

Table 6: Few-shot and zero-shot generated queries randomly sampled from ArguAna, FiQA and HotpotQA dataset.

## C  DETAILED IMPLEMENTATION

Figure 5 shows the overall process of PROMPTAGATOR++, the details of which are in Section 3.

Our cross attention reranker is a listwise model based on T5. Specifically, it takes a list of documents given a query as the input. We represent each query-document pair as "`Query: {q} Document: {d}`" and feed it into the encoder of a T5 model. We then apply a projection layer on the output encodings of the first token and use the output as the ranking score. We optimize the model using softmax cross entropy loss over over a ranking list consisting of a positive $(q, d^+)$ pair and 31 sampled negative $(q, d^-)$ pairs. Unlike monoT5 (Nogueira et al., 2020), which is pointwise reranker that uses an encoder-decoder model and is trained to generate a relevance label, our model is a listwise reranker and is directly optimizes for ranking performance.

## D  QUERY GENERATION STATISTICS

In Table 7, we analyze the length of the generated questions by different query generation systems. Note that NQ-QGen always generates short queries due to the query generation models being fine-tuned on the NQ dataset, and all of the generated questions have similar length to those questions of NQ. Interestingly, zero-shot PROMPTAGATOR already obtains more variance in terms of length compared to NQ-QGen. Finally, few-shot PROMPTAGATOR offers significantly more variance in terms of the length of generated queries.

| | Few-shot | Zero-shot | NQ QGen |
|---|---|---|---|
| ArguAna | 98.2 | 26.0 | 9.7 |
| Touché-2020 | 7.8 | 13.4 | 9.8 |
| TREC-Covid | 10.8 | 11.4 | 10.2 |
| NFCorpus | 8.3 | 11.5 | 10.3 |
| HotpotQA | 11.2 | 12.2 | 8.8 |
| DBPedia-Entity | 8.2 | 13.8 | 8.8 |
| Fever | 12.1 | 10.7 | 8.8 |
| Climate-Fever | 12.9 | 10.7 | 8.8 |
| SciFact | 12.6 | 12.4 | 10.0 |
| SCIDOCS | 7.4 | 15.7 | 10.7 |
| FiQA-2018 | 12.5 | 10.1 | 9.5 |
| AVG. | 17.8 | 13.5 | 9.6 |

Table 7: Average query length.

## E  ROUND-TRIP FILTERING EXAMPLES

Figure 6 shows examples of queries from few-shot PROMPTAGATOR that were removed by round-trip filtering.

## F  ROUND-TRIP FILTERING AS LATENT VARIABLE MODELING

This section aims to share some insight into why our round-trip filtering method is effective, by viewing queries as latent variables that we estimate. First, consider a hypothetical graphical model in which each query $q$ is a latent variable and the documents retrieved for that query are observed variables following some distribution, $p(d|q, \theta^*)$, where $\theta^*$ represents the parameters of a hypothetical "optimal" retriever that always selects the "best" documents for any query. For synthetic data generation, we make it our goal to sample queries from the posterior, $p(q|d, \theta^*)$, which according to Bayes rule is:

$$p(q|d, \theta^*) = \frac{p(d|q, \theta^*)p(q|\theta^*)}{\sum_{q'} p(d|q', \theta^*)p(q'|\theta^*)}$$

where $p(q|\theta^*)$ is a prior over queries. We will assume it is an "uninformative prior" that is uniform over all $q$ and later write it as just $p(q)$. If we knew $\theta^*$ (the parameters of an optimal retriever), we could just directly compute the above expression. But in practice we do not, so we will estimate $\theta$ using expectation maximization (EM) (Dempster et al., 1977), which will turn out to mirror our round-trip filtering algorithm. EM and other latent variable learning methods have long been used to impute missing data with great success.

In EM, we estimate $\theta$ by approximately maximizing the marginal likelihood of the observed variables:

$$\hat{\theta} = \arg\max_{\theta} \prod_{d \in \mathcal{D}_T} p(d) = \prod_{d \in \mathcal{D}_T} \sum_q p(d|q, \theta)p(q)$$

The first step in EM is to make an initial estimate of $p(q|d)$ for every document $d$. We use our FLAN query generator as our initial estimate: $p_{\text{FLAN}}(q|d)$. We then proceed to the M-step of EM, which computes:

$$\hat{\theta} = \arg\max_{\theta} \sum_{d \in \mathcal{D}_T} \sum_{q} p_{\text{FLAN}}(q|d)p(d|q, \theta)$$

This is equivalent to training an initial retriever $p(d|q, \hat{\theta})$ on documents from $\mathcal{D}_T$ that have been paired with our FLAN-generated queries (what we do). Finally, we proceed to the E-step of EM, which estimates $p(q|d, \hat{\theta})$ for every document $d$:

$$p(q|d, \hat{\theta}) \propto p(d|q, \hat{\theta})p(q)$$

From this, we see that the highest probability queries under $p(q|d, \hat{\theta})$ are the ones with the highest probability of retrieving $d$ under our initial retriever (since $p(q)$ is an uninformative prior that is uniform over all $q$).

To sample from $p(q|d, \hat{\theta})$, we could employ importance sampling. In importance sampling, we first sample $q$ from any proposal distribution that we choose, $p_{\text{prop}}(q|d)$. We would then weight that sample by the ratio $p(q|d, \hat{\theta})/p_{\text{prop}}(q|d)$. We choose $p_{\text{FLAN}}(q|d)$ as our proposal distribution. Then, instead of actually importance-weighting each sample, our filtering procedure just discards samples with a low value of $p(d|q, \hat{\theta})$, which is proportional to the numerator in the importance weight. This reveals both the differences and connections between our method and EM.

Although we could repeat this EM-like procedure until convergence, we found that a single round of filtering yielded sufficient quality gains. It is also worth noting that the training of our final retriever could be viewed as another M-step.

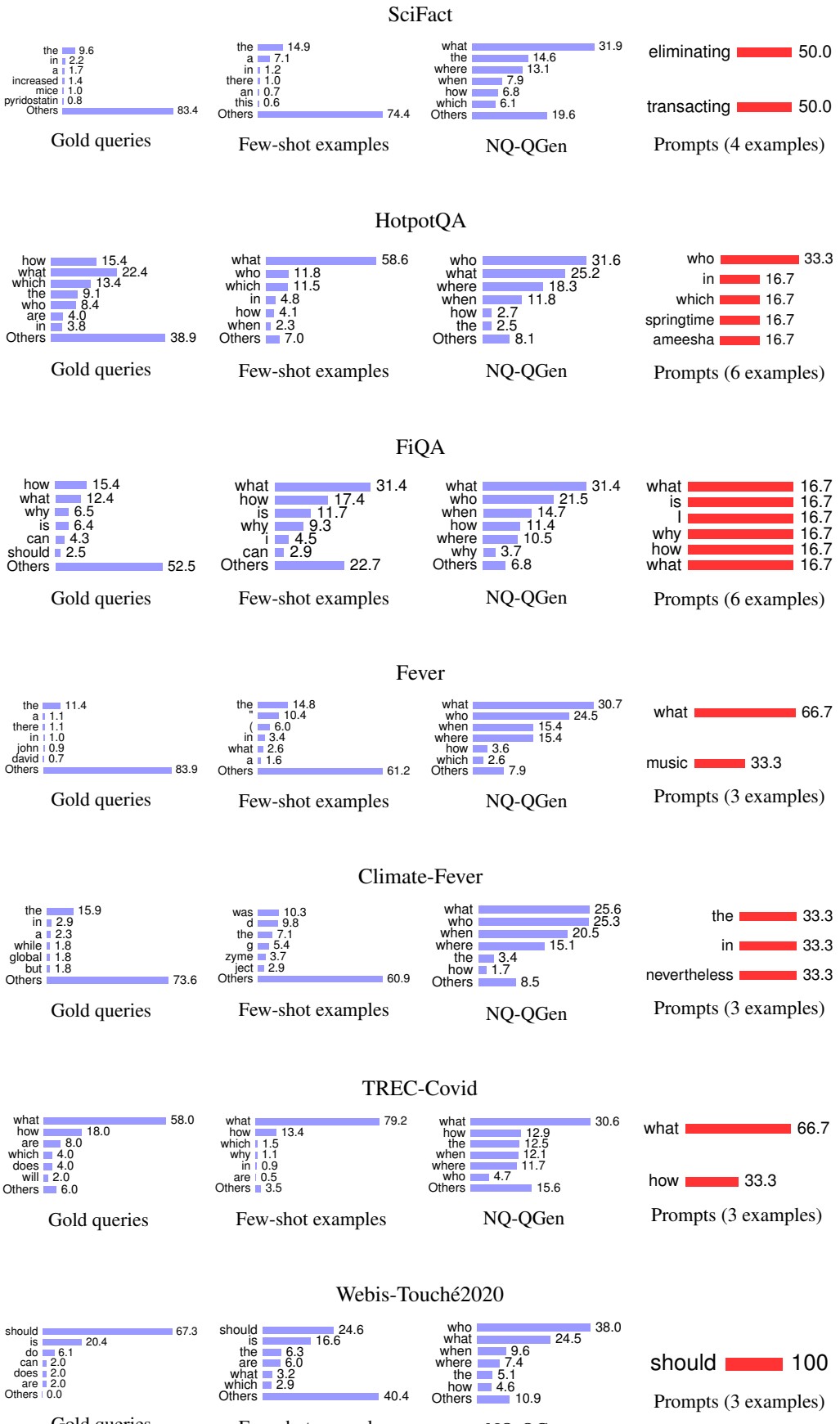

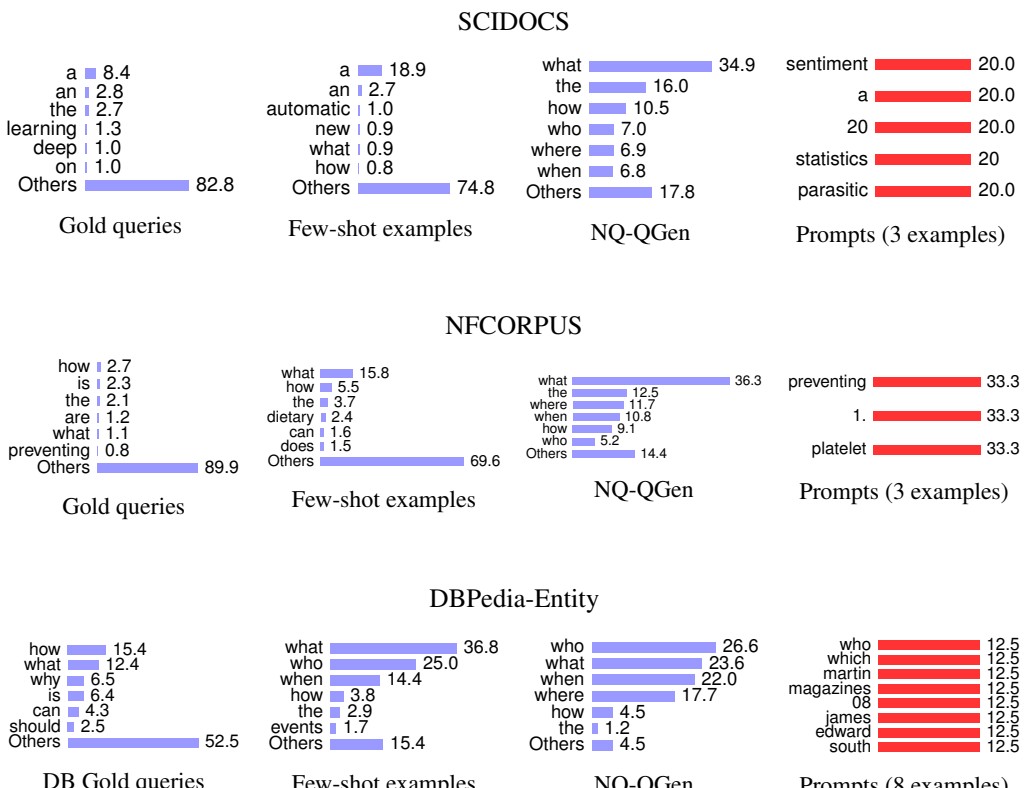

Figure 4: Top first word distribution on queries generated from different models in all other BEIR datasets.

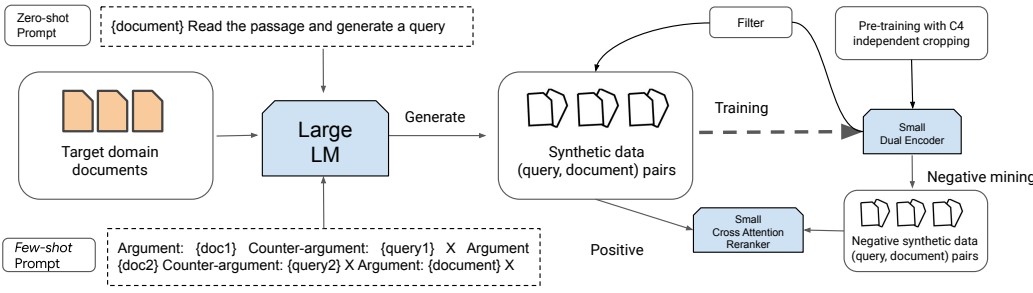

Figure 5: PROMPTAGATOR++ Training pipeline.

**Passage:** As the COVID-19 pandemic sweeps the globe, evolving containment measures have created an unprecedented need for rapid and effective science communication that is able to engage the public in behavioural change on a mass scale. Public health bodies, governments, and media outlets have turned to comics in this time of need and found a natural and capable medium for responding to the challenge...
**Query:** What is the authors' purpose for writing this article?
**Remarks:** The query lacks context. This query is also generic and can be used to all articles

**Passage:** Big Bad Love is a 2001 film directed by Arliss Howard , who co-wrote the script with his brother , James Howard , based on a collection of short stories of the same name by Larry Brown . The story recounts an episode in the life of an alcoholic Vietnam veteran and struggling writer named Leon Barlow , who is played by Arliss Howard , and his wife , played by Howard 's wife Debra Winger . The soundtrack includes music by Tom Verlaine , the Kronos Quartet , and R. L. Burnside .
**Query:** music artist is from the United States
**Remarks:** The query lacks context.

**Passage:** Aminopeptidase N (APN) is the major cell surface receptor for group 1 coronaviruses. In this study, we have isolated and characterized a feline APN cDNA and shown that the transfection of human embryonic kidney cells with this cDNA renders them susceptible to infection with the feline coronavirus feline infectious peritonitis virus, the human coronavirus (HCV) 229E and the porcine coronavirus porcine transmissible gastroenteritis virus. ...
**Query:** What is the function of Aminopeptidase N?
**Remarks:** The query is too general and can match many documents that mentions APN.

**Passage:** Selena Danielle Coppa (born February 25, 1983) was a military intelligence Sergeant in the United States Army. She is primarily notable for her organizing and activism against the US Occupation of Iraq while serving as an active duty military member, including serving on the Executive Board of Iraq Veterans Against the War. In 2009 it was announced that she was heading a committee responsible for gaining and training more active duty anti-war soldiers.
**Query:** who was the first president of the senate?
**Remarks:** Hallucination. The passage didn't mention president of the senate.

**Passage:** Porcine deltacoronavirus (PDCoV) is an emerging swine coronavirus that causes severe diarrhea, resulting in high mortality in neonatal piglets. Despite widespread outbreaks in many countries, no effective PDCoV vaccines are currently available. Here, we generated, for the first time, a full-length infectious cDNA clone of PDCoV. We further manipulated the infectious clone by replacing the NS6 gene with a green fluorescent protein (GFP) to generate rPDCoV-NS6-GFP...
**Query:** What virus is the first one that we've been able to create a vaccine for?
**Remarks:** Hallucination. The passage didn't mention that Porcine deltacoronavirus is the first virus one can create a vaccine.

**Passage:** Nacoleia rectistrialis is a moth in the Crambidae family. It was described by Hampson in 1912. It is found on the Key Islands.
**Query:** Who composed the music for the movie the ring?
**Remarks:** Hallucination. The passage is about moth and is unrelated to "music for the movie the ring".

Figure 6: Examples of queries from few-shot PROMPTAGATOR's prompt-based query generation that were removed by round-trip filtering.

