# OpenReview forum: "Promptagator: Few-shot Dense Retrieval From 8 Examples"
_ICLR.cc/2023/Conference — ICLR 2023 poster_

### Official Review · Reviewer_FSYW · 2022-10-20

**Confidence:** 4
**Correctness:** 3
**Technical Novelty And Significance:** 2
**Empirical Novelty And Significance:** 3
**Recommendation:** 5

**Clarity, Quality, Novelty And Reproducibility:**

This work is clearly presented and well-written. The idea of using LLM to generate synthetic data for model training is not new, even for the retrieval task (UPR, InPars), but the authors claim that their approach differs in that i) they do task-specific adaptation, and ii) they focus on first-stage retrieval instead of (or in addition to) reranking.

The method should be relatively straightforward to reproduce theoretically, if one has the required computation power. In practice, very few would be able to query a 100B LLM to generate millions of samples for their retrieval task.

**Strength And Weaknesses:**

Strengths:
- The argument that retrieval tasks differ in "search intents" and "query distributions" is sound and worth considering when designing a retriever in the low-resource setting.
- The proposed method is straightforward and clearly presented.
- As a general data augmentation approach, (theoretically) the proposed method can also be used to train other retrievers.

Weaknesses:
- In order to use the proposed approach in practice, for each target retrieval task, one needs to query a 100B LLM to generate millions of samples, which is impractical for almost everyone but a few who have access to an enormous amount of computation power.
- In this work, a separate retriever is trained for each target retrieval task. Compared to the traditional "transfer learning" setting where a single retriever is trained to generalize to many new target tasks (in either zero-shot or few-shot fashion), the proposed paradigm is arguably more cumbersome and expensive. For instance, to address the concerns of different "search intents" and "query distributions", it is potentially better to train a general multi-task retriever that can be easily adapted to new target tasks with different intents using prompts (instructions) or other techniques.
- As the proposed setting is different from virtually all previous works, the comparison is not exactly apple-to-apple. For instance, this work utilizes the target corpus during training (to generate queries), which is not the case for many baseline methods. Also this method is training a separate retriever for each target task, effectively increasing the model capacity, which is also different from the baselines.

**Summary Of The Paper:**

This paper focuses on low-resource text retrieval setting that is different from the more widely considered settings so far. In particular, to address the lack of annotated data for retrieval tasks, existing works usually aim to train a retriever using available labeled data (e.g. MS MARCO) with good generalization to a range of target tasks (e.g. BEIR) in a zero-shot fashion. This work, instead, tries to build a task-specific retriever for each target task using a handful of target-task annotations (few-shot).

To achieve this end, they prompt a very large LM (the 137B FLAN) with the few-shot annotations, as well as the entire document corpus, from the target retrieval task to generate millions of pseudo-queries. These pseudo query-document pairs are then filtered using a retriever to improve their quality. Eventually, the filtered synthetic query-document pairs can serve as training data to train i) a bi-encoder neural retriever and (optionally) ii) a cross-encoder reranker. This proposed model is evaluated on a number of retrieval tasks from BEIR which shows competitive results.

**Summary Of The Review:**

This paper identifies a valid issue in low-resource retrieval, which is the difficulty of capturing the diverse "search intents" and "query distributions" in different target tasks, but the proposed task-specific few-shot setting is, in my opinion, less than ideal compared to the existing transfer learning paradigm. It proposes a straightforward method to generate synthetic data with LLMs and achieves competitive results on BEIR, although the comparison with baselines is not exactly apple-to-apple. Finally, the proposed approach is very expensive in practice, and may not be well justified given its performance improvement over other retrievers that are simply trained on MS MARCO and adapted to the target dataset in a zero-shot fashion.

---

> ### Author Response · Authors · 2022-11-19
> **Response to Reviewer FSYW**
>
> > ***[1]*** one needs to query a 100B LLM to generate millions of samples
>
> Generating synthetic data with LLM is costly right now, but we believe the value of our work is to demonstrate the potential of using LLMs.  It is also worth noting that there are several improvements on the training and serving large language models such as intrustruction tuning (e.g. FLAN-T5, 13B) , more efficient tuning (e.g., chinchilla), and cheaper inference (e.g. LLM.int8), and potentially the inference cost of LLM can get significantly cheaper overtime.  (The price of GPT-3 has decreased by 3x over the past year from 0.06/1K tokens to 0.02/1k tokens).
>
> > ***[2]*** it is potentially better to train a general multi-task retriever that can be easily adapted to new target tasks with different intents using prompts (instructions) or other techniques.
>
> Thanks for the suggestion. Multi-task learning for diverse retrieval tasks like BEIR is an interesting but non-trivial research problem. It remains underexplored in prior work– previous SOTAs on BEIR used single task training, or task-specific data augmentation (QGen) like our work. The open questions, to name a few, include how to collect diverse training data (multi-task DPR tasks are solely on wikipedia), and how much the dual encoder architecture can generalize.  Due to the time constraints, we will leave this to future work.
>
> > ***[3]*** this work utilizes the target corpus during training (to generate queries), which is not the case for many baseline methods.
>
> Many baseline methods in BEIR do utilize the target corpus to form task-specific retrievers. In fact, all of the query generation baselines such as GenQ and GPL generated synthetic questions on the target corpus and trained task-specific retrievers. Even for BM25, different idf values are computed using different target corpuses, resulting in different task-specific retrievers.
>
> > ***[4]*** As the proposed setting is different from virtually all previous works, the comparison is not exactly apple-to-apple.
>
> The central idea of this paper is to showcase that every retrieval task is different, and a few examples is enough to learn a good task-specified retriever. On that front, our setting is already different from prior work.  To add more baselines, we have added the study of including the 8 examples into the directly fine-tuning of each task on top of GTR dual encoder baseline (base model, 110M). To use the 8 examples more effectively, we associated each positive example with 32 random negative examples from the target corpus to construct a batch, and ran 50 steps of fine-tuning. As expected, the 8 examples did not provide a large amount of impact for GTR base, where the average nDCG go from 40.4 to 38.7 after 8 examples fine-tuning.
>
> | Model                         |  arg | touche | covid | nfc | hotpot | dbp | climate | fever | scifact | scidocs | fiqa | *AVG* |
> | ------------------------- | --- | ------ | ---- | --- | --- | --- | ------------ | ------ | ------- | ---------- | ---- | ------ |
> | GTR-base                   |  51.1 | 20.5 | 53.9 | 30.8 | 53.5 | 34.7 | 24.1 | 66.0 | 60.0 | 14.9 | 34.9 | *40.4* |
> | GTR-base + 8-shot ft |  51.9 | 23.9 |  56.2 | 30.5 | 23.4 | 35.4 | 25.4 | 68.2 | 60.1 | 14.8 | 35.3 | *38.7* |

---

> ### Author Response · Authors · 2022-11-27
> **Looking forward to your feedback**
>
> Dear reviewer,
>
> We hope that you've had a chance to read our responses. We would really appreciate a reply as to whether our responses and clarifications have addressed the issues raised in the review, or whether there is anything else we can address.

---

### Official Review · Reviewer_Qvy4 · 2022-10-23

**Confidence:** 4
**Correctness:** 3
**Technical Novelty And Significance:** 3
**Empirical Novelty And Significance:** 3
**Recommendation:** 6

**Clarity, Quality, Novelty And Reproducibility:**

Some in-the-details questions
- in section 4.1, reranking the top 200 candidates. Is that standard? Do other works use 200 as well? For each task what % of the time was the correct document in the top 200?
- was the pretraining the retriever on the C4 data the same as Contriever? (Apologies, I haven't gone over that work). If so, why was this done, rather than starting with the Contriever model and fine-tuning that?

**Strength And Weaknesses:**

Strengths I saw:
* Most details are clearly described on the proposed approach and the conducted experiments.
The results are on public datasets, and public "foundation model" checkpoints.

* The argument that there are different types of retrieval problems, where for one task (queryA, docA) is a valid pair, and for another it isn't, is well motivated.

* The results on using the initially trained retriever model, to then score the synthetic data it was being trained on, remove anything not meeting adequate confidence that (query, doc) is a valid pair, and then continue training is a nice result.

* The strengths of the zero-shot data gen results are also promising.

Regarding weaknesses, I don't see any red flags that question the results or merit of the work. I do have some questions though.
* The first is around Promptagator using FLAN as it's LLM , compared to (Thakur et al. 2021) and this papers own use of T5 as the question generation model. The results in table 3 hint that if prior works had used FLAN rather than T5, GPT etc for their Q generation, then maybe their results would be at Promptagators, or better. Also, why is the "FLAN original" result in table 3 not one of the rows in table 2?
* Is the Intent(doc, query) premise (raised in the intro about how there are different types of retrieval tasks) not weakened by the rows in Table 2 where pre-training on MSMARCO results in good gains?

* Minor, but some of the discussion where the fact the model is smaller than others (125M params) is slightly misleading I thought, since a 139B param model was used for data synthesis. Point completely agreed on inference time though.

**Summary Of The Paper:**

The paper presents an approach to using a large language model (LLM) to synthesize fine tuning data for retrieval. Specifically, a question is generated for a given document, where the document is "answers" the question. "Answers" can have a task specific meaning, hence the quotation marks. A small number of task specific examples are used as prompts in the generation of this question. This synthetic data is then used to fine tune a retrieval model (Promptagator, scores query and candidate document, independent of other docs), or a retrieval model plus cross-attention reranker model (Promptagator++, ie a model that models some relation between query and candidate documents).
The paper presents results across 10 retrieval tasks (datasets), showing that this approach results on average in accuracy gains, according to the retrieval @10 metric.

The paper is well written. It must be said that it combines and builds upon existing ideas and very related work, but it is likely of value and interest to the retrieval community.

**Summary Of The Review:**

A solid, iterative contribution to the document retrieval literature.

---

> ### Author Response · Authors · 2022-11-19
> **Response to Reviewer Qvy4**
>
> >  ***[1]*** … The results in table 3 hint that if prior works had used FLAN rather than T5, GPT etc for their Q generation, then maybe their results would be at Promptagators, or better….
>
> We believe our promptagator zero-shot result already covers this experiment; it is similar to GenQ in (Thakur et al. 2021), performing query generation for different tasks, but using FLAN instead of T5. We verified that
> FLAN is more powerful than T5. However, the few-shot prompt adds 2.3 nDCG on promptagator and 2.9 nDCG on promptagator+, showing the importance of few-shot prompting.
>
> We show FLAN original in Table 3 separately to declutter the main table, given that there are four variations of promptagator: (zero-shot, few-shot) x (promptagator, promptagator+).
>
> > ***[2]*** Is the Intent(doc, query) premise (raised in the intro about how there are different types of retrieval tasks) not weakened by the rows in Table 2 where pre-training on MSMARCO results in good gains?
>
> Pretraining on out-of-domain data like MSMARCO/NQ does give gains to our baselines. Surprising, for promptagator, we found it worse than pretraining with the unsupervised Contriever objective (https://arxiv.org/pdf/2112.09118.pdf). We suspect that when the generated data quality is good, adding out-of-domain training data might not always help the model.
>
> > ***[3]*** 139B param model was used for data synthesis.
>
> Thanks for pointing this out. We have added a footnote in the comparison table to make this point clear.
>
> > ***[4]*** ...reranking the top 200 candidates. Is that standard?
>
> Most work rerank the top 1000 candidates in the BEIR benchmark such as monoT5 and MiniLM (https://arxiv.org/pdf/2206.02873.pdf). We choose to rerank 200 as it is much faster and also performs slightly better for our model.
>
> > ***[5]*** Was pretraining the retriever on the C4 data the same as Contriever?
>
> Contriever was trained on Wiki and CCNet, and we train our own version on C4.  The primary reason for creating our own version is checkpoint compatibility, since our dual encoder models are based on T5 and Contriever is based on BERT.

---

> ### Author Response · Authors · 2022-11-27
> **Dear reviewer,**
>
> Dear reviewer,
>
> We hope that you've had a chance to read our responses. We would really appreciate a reply as to whether our responses and clarifications have addressed the issues raised in the review, or whether there is anything else we can address.

---

### Official Review · Reviewer_HWjD · 2022-10-24

**Confidence:** 5
**Correctness:** 4
**Technical Novelty And Significance:** 3
**Empirical Novelty And Significance:** 3
**Recommendation:** 8

**Clarity, Quality, Novelty And Reproducibility:**

Clarity&Quality: The paper is well-written and easy to follow
Novelty: Good novelty -- new task setting was developed.
Reproducibility: The authors didn't provide the code so we cannot reproduce it.

**Strength And Weaknesses:**

Strength:
- The authors proposed a new few-shot setting for the BEIR benchmark, which is more reasonable because of the wide discrepancy between all the tasks in BEIR.
- With only 8 examples, PROMPTAGATOR is able to outperform supervised models that are trained on labeled paired datasets.
- Baselines and references of prior work are sufficient for comparison.
- Round trip filtering is designed and evaluated as a good method to clean the dataset for better performance.

Weaknesses:
- For all the baseline models you have for BEIR, they are not adapted with the 8 examples. So it's basically not a fair comparison. I know 8 examples may not help a lot for these baseline models, but I think it's better to add the results by adapting the models using 8 examples.

**Summary Of The Paper:**

The author proposed a few-shot retrieval evaluation setting, which addresses the difference in the search intent and query distribution for different retrieval tasks. They also proposed a simple recipe for few-shot retrieval by prompting an LLM to generate synthetic task-specific training data and then train a dual encoder on the generated dataset. The experiment results show that its few-shot performance with two-to-eight examples is even better than a fully supervised model that is trained on MS MARCO or NQ.

**Summary Of The Review:**

This paper proposed (1) a new task setting for BEIR retrieval benchmark (2) a few-shot method for learning a retrieval model by prompting the LLMs to generate data. The results are solid. I would like to recommend this paper be accepted.

---

> ### Author Response · Authors · 2022-11-19
> **Response to Reviewer HWjD**
>
> > For all the baseline models you have for BEIR, they are not adapted with the 8 examples. So it's basically not a fair comparison. I know 8 examples may not help a lot for these baseline models, but I think it's better to add the results by adapting the models using 8 examples.
>
> Thanks to the reviewer for the suggestions! We have added the study of including the 8 examples into the fine-tuning of each task on top of GTR dual encoder baseline (base model, 110M). To use the 8 examples more effectively, we associated each positive example with 32 random negative examples from the target corpus to construct a batch, and ran 50 steps of fine-tuning. As expected, the 8 examples did not provide a large amount of impact for GTR base, where the average nDCG goes from 40.4 to 38.7 after 8 examples fine-tuning.
>
> | Model                         |  arg | touche | covid | nfc | hotpot | dbp | climate | fever | scifact | scidocs | fiqa | *AVG* |
> | ------------------------- | --- | ------ | ---- | --- | --- | --- | ------------ | ------ | ------- | ---------- | ---- | ------ |
> | GTR-base                   |  51.1 | 20.5 | 53.9 | 30.8 | 53.5 | 34.7 | 24.1 | 66.0 | 60.0 | 14.9 | 34.9 | *40.4* |
> | GTR-base + 8-shot ft |  51.9 | 23.9 |  56.2 | 30.5 | 23.4 | 35.4 | 25.4 | 68.2 | 60.1 | 14.8 | 35.3 | *38.7* |

---

> > ### Comment · Reviewer_HWjD · 2022-11-27
> > **Thanks for the reply**
> >
> > Thanks for the reply and the additional experiments. I will keep my score unchanged.

---

### Official Review · Reviewer_ZHw5 · 2022-10-31

**Confidence:** 4
**Correctness:** 4
**Technical Novelty And Significance:** 3
**Empirical Novelty And Significance:** 3
**Recommendation:** 8

**Clarity, Quality, Novelty And Reproducibility:**

New setting, simple and novel method, clear writing. Would like to see more baselines.

**Strength And Weaknesses:**

Strengths:
* Novel and semi-practical setting of few-shot retrieval in novel domains.
* Simple and also very timely solution (instruction-tuned LLMs)
* Sensible evaluation, common datasets and pretty sensible baselines.
* Paper is clear and well-written

Weaknesses:
* Would have loved to see more multitask baselines. In particular, part of the paper's motivation is that rather than multi-tasking and hoping to generalize to the target task, one would like to quickly adapt to the target distribution. In that sense, it'd be interesting to see some methods such as training a multi-task biencoder and running it zero-shot on the target domain, or finetuning that multi-task biencoder on the few-shot examples. (I am thinking of the multi-task DPR model from Maillard et al. 2021).
* Not actually instruction-tuning a QA-specific model. Instead, the method is basically just querying off-the-shelf systems.

Random Idea:
* It would be an interesting idea to see how well you could use LLMs for retrieval without needing to train a seperate retriever (although it would be incredibly slow). In particular, take a prompted LLM that takes as input a series of question-paragraph pairs, and then score test paragraphs using p(paragraph|question). The reason I think this is interesting is because in your current method, you are essentially distilling a large LM into a dual encoder by generating synthetic data. I am wondering if doing the distillation causes the dual encoder to actually be a worse or better retriever than the original slow LM.

**Summary Of The Paper:**

This paper studies the problem of training a few-shot dense retriever, i.e., a dual encoder model that can operate well in on a new distribution with only a small number of question-paragraph pairs. To do so, they train a large LM to generate questions given unlabeled paragraphs as input. This LM is trained in a instruction-tuned way, and then prompted at test time with the few examples of the desired task. After generating the synthetic question-paragraph examples, they train a seperate dual encoder model on the data.

**Summary Of The Review:**

I think the paper is compelling enough to warrant an accept given its clear writing, new methods, and sensible evaluation.

---

> ### Author Response · Authors · 2022-11-19
> **Response to Reviewer ZHw5**
>
>  >  ***[1]*** Would have loved to see more multitask baselines.
>
>  Thanks for the suggestion. Multi-task learning for diverse retrieval tasks like BEIR is an interesting but non-trivial research problem. It remains underexplored in prior work– previous SOTAs on BEIR used single task training, or task-specific data augmentation (QGen) like our work. The open questions, to name a few, include how to collect diverse training data (multi-task DPR tasks are solely on wikipedia), and how much the dual encoder architecture can generalize.
> In addition, we believe that promptagator and multi-task learning are complementary in the following regards: the multi-task model needs to be trained on the union of datasets from all tasks, and learns a single model.  On the other hand promptagator can generate data for low resource tasks,  which can provide a large set of training data that spans a wide range of tasks for multi-task learning. Meanwhile, given a multi-task model, one can further adapt it to a new task using promptagator.
>
> > ***[2]*** Not actually instruction-tuning a QA-specific model. Instead, the method is basically just querying off-the-shelf systems.
>
> In this work, we want to highlight that using an off-the-shelf LLM already enables few-shot learning of dense retrievers; this is important because finetuning LLMs is expensive. But there is headroom for instruction-tuning a specific model to improve query generation quality or efficiency. This is worth exploring in future work.
>
> > ***[3]*** It would be an interesting idea to see how well you could use LLMs for retrieval without needing to train a seperate retriever (although it would be incredibly slow).…
>
> This is a good idea. We have the UPR baseline that uses a similar approach – it ranks documents based on T0’s p(question|paragram) at the reranking stage. However, it is unclear how to run such an approach efficiently for the retrieval stage, where we need to score O(Million) documents for each question at serving time. Having a model that has the LLM capability but also allows efficient retrieval is definitely worth exploring.

---

### Author Response · Authors · 2022-11-19
**General Response**

Thanks to all reviewers for the detailed reviews and suggestions.  In the latest version, we updated the paper to include another baseline result by fine-tuning GTR dual-encoder using the few-shot examples directly. We address each reviewer's comments separately below.

---

### Public Comment · ~Cheng-Han_Chiang1 · 2023-04-12
**Plan to Release the Source Code and Models**

Dear authors,
Thank you for this interesting work. I am wondering if there are any plans to release the source code, models (e.g., the unsupervised pre-trained retriever), and the 8-shot query-document pairs that are used in the paper.

---

### Decision · Program_Chairs · 2023-01-20

**Decision:**

Accept: poster

**Justification For Why Not Higher Score:**

The idea is simplistic. It would have got a higher score if the paper has more technical depth. Many reviewers mention the multitask setting. The authors also agree that this is an interesting direction, but decides to limit the scope of the paper.

**Justification For Why Not Lower Score:**

The idea is simple and effective.

**Metareview: Summary, Strengths And Weaknesses:**

This paper proposes to use large language models to generate queries for a small set of documents in a few shot retrieval setting. The setting itself is part of the novelty. The idea is simple and practical. Since this is a new setting, the focus should be on the evaluation of the baseline and the improvement over the baseline. There are some discussion about whether the baseline is strong enough and whether it is fair for the baseline. Most reviewers (including me) are convinced.

There is a concern about whether using FLAN takes up too much compute. It is indeed a concern given that the paper has claims about the low compute it needs. The wording can be fixed. The compute problem will be fixed as GPUs get more powerful, so overall I would OK with this issue.

**Note From Pc:**

if the above contains the word "oral" or "spotlight" please see: "oral" presentation means -> notable-top-5% and "spotlight" means -> notable-top-25%. As stated in our emails, we are disassociating presentation type from AC recommendations

**Summary Of Ac-Reviewer Meeting:**

Two of the reviewers engaged in the discussion. No red flags raised in the exchange.